# Every Apple has a Voice: Using Stable Isotopes to Teach about Food Sourcing and the Water Cycle

Erik Oerter[1,2,4], Molly Malone[3], Annie Putman[1,2], Dina Drits-Esser[3], Louisa Stark[3], Gabriel Bowen[1,2]

[1] Department of Geology and Geophysics, University of Utah, 115 South 1460 East, Salt Lake City, UT 84112, USA.

[2] Global Change and Sustainability Center, University of Utah, 257 South 1400 East, Salt Lake City, UT 84112, USA.

[3] Genetic Science Learning Center, University of Utah, 515 100 South, Salt Lake City, UT 84112, USA.

[4] Current address: Lawrence Livermore National Laboratory, 7000 East Avenue, Livermore, CA 94550, USA.

*Correspondence to*: Erik Oerter (erikjoerter@gmail.com)

**Abstract.** Agricultural crops such as fruits take up irrigation and meteoric water and incorporate it into their tissue ("fruit water") during growth, and the geographic origin of a fruit may be traced by comparing the H and O stable isotope composition ($\delta^2$H and $\delta^{18}$O values) of fruit water to the global geospatial distribution of H and O stable isotopes in precipitation. This connection between common fruits and the global water cycle provides an access point to connect with a variety of demographic groups to educate about isotope hydrology and the water cycle. Within the context of a one-day

outreach activity designed for a wide spectrum of participants (high school students, undergraduate students, high school science teachers) we developed introductory lecture materials, in-class participatory demonstrations of fruit water isotopic measurement in real time, and a computer lab exercise to couple actual fruit water isotope data with open-source on-line geospatial analysis software. We assessed learning outcomes with pre- and post-tests tied to learning objectives, as well as participant feedback surveys. Results indicate that this outreach activity provided effective lessons on the basics of stable

isotope hydrology and the water cycle. However, the computer lab exercise needs to be more specifically tailored to the abilities of each participant group. This pilot study provides a foundation for further development of outreach materials that can effectively engage a range of participant groups in learning about the water cycle and the ways in which humans modify the water cycle through agricultural activity.

**Keywords**

Science education, Isotope hydrology, virtual water footprint, irrigation, water transfers, soil water, forensics, food sourcing

# 1 Introduction

Teaching of environmental science, and enhancing student engagement with associated disciplines is a key focus of science education efforts in the United States and worldwide (Quinn et al., 2012). Because human activity and climate change combine to create new pressures on water resources (Pachauri et al., 2014), a comprehensive understanding of how water moves between reservoirs in Earth's subsurface, surface and atmosphere ('The Water Cycle'; e.g. Oki and Kanae, 2006) is critical knowledge for students in any field of study. The economic and ecological implications of changes to the water cycle provides motivation for increased focus on environmental sciences and improvement in the teaching of these sciences in K-12 and undergraduate science education curricula in the United States and internationally (Osborne and Dillon, 2008; Quinn et al., 2012; Rudell and Wagener, 2013; Wagener et al., 2007). However, the invisibility of some components of the Water Cycle, such as inter-reservoir fluxes (e.g., atmospheric water vapor transport) is a fundamental challenge in teaching the concepts of the global water cycle (Bar, 1989; Rappaport, 2009). Anthropogenic forcings, such as transfers between watersheds in the form of water diversions for irrigation and drinking water (Good et al., 2014), further complicate the already-complex natural system.

Students' most direct connections to the water cycle are the water they drink and the food they eat. These existing connections may be a novel entry point for education. Existing studies of food provenance have been forensic studies of counterfeiting and ingredient substitutions (Cerling et al., 2015). This present work relies on the fact that agricultural crops such as fruits and vegetables take up precipitation and irrigation water during growth and incorporate it into their tissue ("fruit water") (Kramer and Boyer, 1995), which may have a specific geochemical signature in the form of its hydrogen and oxygen isotopic composition (detailed below). Therefore, if the fruit water can be linked to its source water, the geographic origin of the fruit may be determined, and the complex relationship between agriculture, transportation of crops to market, and consumption becomes more apparent.

The distance between food source and consumption represents the movement of water out of source watersheds via trucks and trains, rather than inside pipes, a concept known as a 'Virtual Water Footprint', which is another nearly-invisible water transport mechanism (Hoekstra and Mekonnen, 2012; Chen and Chen, 2013). For example, in California in 2015, 2.3E+10 kg of vegetable crops were harvested and transported to market (CDFA, 2016), which represents 2.17E+10 L of water (average vegetable water content is 95% (Spungen, 2005)). While this vegetable water is two orders of magnitude smaller than the volume of water transported for urban uses in California in 2015 (7E+12 L (Mount and Hanak, 2016)), the vegetable crop water is an enormous amount of water being moved around the landscape that many people may not have considered.

We designed a one-day teaching activity within the context of a week-long outreach workshop that was centered on helping aspiring science, technology, engineering, and math (STEM) students and high school science teachers gain insight into the practice of science and to engage the participants in hands-on hydrological research. For this teaching activity, we developed introductory lecture materials, in-class participatory demonstrations of fruit water isotopic measurement in real

time, and a computer lab exercise to couple actual fruit water isotope data with open-source on-line geospatial analysis software. The goals of the teaching activities were to introduce and establish key water cycle concepts, reveal human modification of the natural water cycle, introduce core concepts of stable isotope hydrology, and link these concepts together through scientific data analysis and synthesis in the context of a food sourcing study. Additionally, because the week-long workshop was conducted as outreach from large teaching and research universities towards ambitious students and teachers, an over-arching goal of the workshop activities (and this paper) was to inspire and nurture scientific interest in the participants by exposing them to a variety of environmental science activities.

In this paper, we describe how we (1) developed teaching modules about isotope hydrology and the water cycle, (2) developed novel laboratory and computer demonstrations and exercises to link water cycle components to the investigation of food sourcing, and (3) evaluate the efficacy of these teaching activities and materials. To do so, we first discuss the scientific background for this teaching activity, and provide a dataset of fruit source water H and O stable isotope values for use in the teaching activities. We then provide and discuss teaching materials in the form of lecture materials, and laboratory exercises and datasets, and provide results and discussion on the evaluation of the lesson's efficacy in achieving learning outcomes. The teaching materials are publicly available to the teaching community for non-profit use, and are included in the accompanying Supplementary Materials.

## 2 Scientific Background and Research Methods

### 2.1 Isotope hydrology

Isotopes are atoms of the same chemical element (defined by the number of protons in the nucleus) that differ in the number of neutrons. In the case of the water molecule ($H_2O$), hydrogen has two stable isotopes (termed 'stable' because they do not undergo radioactive decay): $^1H$ and $^2H$ (where the superscript numeral refers to the number of protons plus neutrons), while oxygen has three stable isotopes: $^{16}O$, $^{17}O$, and $^{18}O$, though $^{17}O$ is present in very low abundance in nature is not included in this work. The heavier isotopes ($^2H$ and $^{18}O$) are more rare in natural systems than their light counterparts ($^1H$ and $^{16}O$), with $^2H$ comprising 0.0156% of all hydrogen atoms, and $^{18}O$ comprising 0.1995% of all oxygen atoms. Because of the low abundance of heavy isotopes, the stable isotopes in a sample are expressed as the ratio of the rare (heavy) isotope to the common (light) isotope in delta ($\delta$) notation as: $\delta = (R_{sample} / R_{standard} - 1)$, where $R_{sample}$ and $R_{standard}$ are the $^2H/^1H$ or $^{18}O/^{16}O$ ratios for the sample and standard, respectively, and values are reported in per mille (‰). The standard for $H_2O$ $\delta^2H$ and $\delta^{18}O$ values is Vienna Standard Mean Ocean Water (VSMOW) (Coplen, 1994).

The $\delta^2H$ and $\delta^{18}O$ values of precipitation (meteoric water) exhibit a linear relationship, termed the Global Meteoric Water Line (GMWL), as shown in Fig. 1, $\delta^2H = 8\ \delta^{18}O + 10$ (Craig, 1961). Precipitation during cold seasons, continental interiors, or in cold regions like high latitudes and elevations tend to exhibit low $\delta^2H$ and $\delta^{18}O$ values, whereas precipitation in warm seasons, at low latitudes and altitudes, and near ocean coasts tend to have higher $\delta^2H$ and $\delta^{18}O$ values (Fig. 1). This results in a global geospatial distribution of $\delta^2H$ and $\delta^{18}O$ values of precipitation that correlates well with mean annual air

temperature (Bowen and Revenaugh, 2003). A detailed discussion of the mechanisms behind these worldwide stable isotope patterns can be found in Bowen (2010).

Because plants take up soil water during growth, the H and O stable isotope composition ($\delta^2$H and $\delta^{18}$O values) of the water in the plant represents the soil water. Therefore, the $\delta^2$H and $\delta^{18}$O values of the plant water can be used to identify plant water sources within the soil (Dawson et al., 2002). By comparing the $\delta^2$H and $\delta^{18}$O values of water measured in fruit to the increasingly well-constrained global geospatial distribution of H and O stable isotopes in precipitation (Bowen and Revenaugh, 2003), the likely growing region of a fruit may be estimated.

A potential caveat of directly linking fruit water to a source region using precipitation is that evaporation may occur from the soil water before plant uptake, after plant uptake and during fruit growth, or after harvest. Water that has undergone evaporation will have higher $\delta^{18}$O and $\delta^2$H values than its source water, and the change in these values proceeds along a linear trend in a plot of $\delta^2$H and $\delta^{18}$O, termed an 'evaporation line' with a slope smaller than that of the GMWL (Fig. 1). Water that has undergone minimal evaporation will plot along the evaporation line but near the GMWL, while progressive evaporation will drive the sample further along the evaporation line and away from the intersection of the evaporation line with the GMWL (Craig, 1961). This model of progressive enrichment through evaporation allows the effects of fruit water evaporation to be detected and corrected for. The simplest approach to this correction involves fitting a regression line through measured $\delta^2$H and $\delta^{18}$O values from multiple samples having the same source water but somewhat different degrees of evaporation (e.g., several pieces of fruit from the same lot but of different size) and back-projecting the regression line to where it intersects with the GMWL (Fig. 1). This intersection of the evaporation line and the GMWL are the source water $\delta^2$H and $\delta^{18}$O values.

## 2.2 Fruit water isotopic analysis

Water $\delta^2$H and $\delta^{18}$O values were measured in the fruits and vegetables listed in Table 1. Hereafter the measurements will be referred to as "fruit water" for brevity, though "vegetable water" is equivalent. Measurements were performed using a membrane-inlet water vapor isotope analyzer system originally developed to sample soil water (Oerter et al., 2017a; Oerter et al., 2017b), and fruit water measurement methods mimic those detailed for soil water in Oerter et al. (2017a). To our knowledge, this is the first application of such a system to measure fruit water $\delta^2$H and $\delta^{18}$O values *in situ*, as well as the use of such a system as a teaching tool.

Briefly, the analytical system consists of a micro-porous membrane probe attached to a laser spectrometer instrument (L2130-i, Picarro, Inc., USA) that simultaneously measures water vapor concentration ([$H_2O$]), $\delta^2$H, and $\delta^{18}$O values in water vapor ($\delta^2H_{vap}$ and $\delta^{18}O_{vap}$) at 1 Hz frequency. A measurement is performed by halving or excavating a small cavity in the fruit item, placing the fruit item in a resealable plastic bag, inserting the membrane probe into the fruit so that it is completely enveloped by the fruit (Figure 2), evacuating the bag of air by squeezing or suction and closing the bag around the probe lines. The fruit water vapor from the probe is carried to the analyzer on dry $N_2$ gas. A measurement duration of 5

minutes is typically sufficient to achieve > 2 minutes of stable [H$_2$O], $\delta^2$H$_{vap}$ and $\delta^{18}$O$_{vap}$ measurements, after which the probe is removed from the fruit and replaced with a clean and dry probe for the next analysis. During each isotopic measurement, a thermometer is inserted into the fruit adjacent to the probe, and fruit temperature is recorded for the duration of isotopic measurement. Because the $\delta^2$H$_{vap}$ and $\delta^{18}$O$_{vap}$ values are measurements of water vapor and not liquid water, they

were converted into liquid water $\delta^2$H and $\delta^{18}$O values ($\delta^2$H$_{liq}$ and $\delta^{18}$O$_{liq}$) by assuming the vapor and liquid were in isotopic equilibrium at the measured sample temperature, using the temperature dependent equilibrium H$_2$O$_{liquid\text{-}water}$ fractionation factors for H and O of Majoube (1971).

It was necessary to measure several ($\geq$ 3) individual fruits of each type in order to characterize the variation in $\delta^2$H$_{vap}$ and $\delta^{18}$O$_{vap}$ values of that fruit type's population. These fruit water $\delta^2$H$_{vap}$ and $\delta^{18}$O$_{vap}$ values were then plotted in $\delta^2$H

and $\delta^{18}$O space and a least-squares regression line was fitted through the fruit water $\delta^2$H$_{liq}$ and $\delta^{18}$O$_{liq}$ values (Figure 3). Each fruit type's source water $\delta^2$H and $\delta^{18}$O values ($\delta^2$H$_{source}$ and $\delta^{18}$O$_{source}$) were then determined by finding the fruit water regression line's intersection with the GMWL. This can be done by either, simply drawing the regression line back to the GMWL intersection (example shown in Figure 4), or by solving the system of equations consisting of the fruit water regression line and the GMWL. An example of this calculation for use in the accompanying laboratory exercise is included

in the Supplementary Materials (Supp. File 2).

### 3 Teaching lesson components

### 3.1 Overview of teaching activities

This teaching and outreach activity took place in the context of a week-long outreach workshop for 9 high school students with an interest in pursuing environmental science, 9 undergraduate students who are actively studying

environmental science and water issues, and 9 high school teachers who teach physical science subjects such as chemistry and earth science. Day one of the workshop consisted of a field-based environmental sampling session where the participants collected water samples from a small creek and documented their sampling activities. Day two was the isotope hydrology activity discussed herein. Day three consisted of collecting responses to a short survey of questions on water-related issues in an urban setting. On day four, the students self-selected into three equally sized groups divided amongst the three days'

concept areas, with the goal of producing a poster exhibit communicating the scientific results of their activities. Each group was guided by a PhD level scientist who specialized in that area of research (and who had guided that day's lesson and sampling activities). Day five consisted of a lunch-time poster exhibition where each group presented their research to a scientific audience. The workshop was designed holistically to expose students to several types of water-related research, but each component-day of the workshop could be a stand-alone activity.

The teaching activities described below consist of three components: 1) An introductory lecture on the water cycle and isotope hydrology general concepts with a participatory demonstration of fruit water isotopic measurements in real time, 2) A computer-based laboratory exercise using the fruit water H and O stable isotope dataset, and 3) Learning effectiveness

evaluations consisting of pre- and post-tests (Supp. File 4) aligned with learning objectives for activities 1 and 2 (Table 2) . The learning objectives are tied to the three dimensions of the Next Generation Science Standards (NGSS Lead States, 2013), which are science curriculum guidelines for grades K–12 in the United States. The NGSS dimensions and the teaching activity associated with each component are: (A) Disciplinary Core Ideas: The Water Cycle and stable isotope hydrology, (B) Science and Engineering Practices: data collection and analysis, (C) Crosscutting Concepts: using data and geospatial relationships to evaluate food sourcing.

A live demonstration of the use of the isotope analyser system to make measurements of fruit water isotopic composition described above and utilizing student participation could be repeated in situations where the instrumentation is available, but is not required. The dataset shown in Table 1 can provide the basis for implementing the following teaching activities and materials in any teaching setting.

### 3.2 Water cycle and isotope hydrology introductory lesson

The first activity consists of a lecture introducing basic water cycle and stable isotope concepts, and linking them with stable isotope hydrology. The content of the Introductory Lesson is focused on teaching to Learning Objectives 1 – 4 (Table 2). The presentation slides used in the teaching session are available for public, not-for-profit use and are provided in the Supplementary Materials. The following discussion in Section 3.2 and 3.3 can be utilized as a teaching guide to accompany the Introductory Lesson materials. Supplementary File numbers referenced in the following discussion refer to slide numbers in the Introductory Lesson presentation slides (Supp. File 1).

The introductory lecture begins with a presentation of the water cycle (Supp. File 1.2). The oceans, atmosphere, and subsurface water are all introduced as "reservoirs" where water resides for some length of time. The processes such as evaporation and runoff associated with movement between reservoirs are introduced as "fluxes". The ways in which human activity and built infrastructure can alter the size of water cycle components are illustrated in Supp. File 1.3.

The chemical formula of the water molecule is introduced in Supp. File 1.3, and Supp. Files 1.4 – 1.7 illustrate how atoms are composed of protons in the nucleus with orbiting electrons, using hydrogen as an example. Isotopes are introduced and illustrated in Supp. Files 1.8 – 1.14, again using hydrogen as the example. The important concept for isotopes is that they are atoms of the same element because they have the same number of protons, but differ in the number of neutrons in the nucleus, with $^1$H and $^2$H as the examples. The central concept for Supp. Files 1.15 – 1.20 is that the differences in how stable isotopes of the same element "behave" in chemical and physical reactions is largely due to differences in mass. In this section, the effects for a water molecule are addressed. Supp. File 1.20 points out the observation that heavier molecules will evaporate less easily than lighter molecules, and this contrasting evaporation behaviour is an important concept for the remainder of the learning activities.

The concept of stable isotope notation is introduced in Supp. Files 1.21 – 1.23, and which can be an obstacle for comprehension and could potentially divert a student's learning progression because the notation is represented by a mathematical equation, the use of a Greek symbol ($\delta$), and the resulting values are all negative numbers (e.g. Orton and

Frobisher, 2004). This is an important juncture to emphasize the simple concept that the notation refers to the ratio of the isotope abundances, and thus lower δ values mean more of the lighter isotopes, and higher δ values mean more of the heavier isotopes, even if all values are negative. More discussion of stable isotope notation as *troublesome knowledge* (i.e. Meyer and Land, 2003; 2005) follows in Section 4.3.

Supp. Files 1.24 – 1.25 show that $\delta^2H$ and $\delta^{18}O$ values in precipitation exhibit substantial geospatial structure and spatial autocorrelation. Supp. Files 1.26 – 1.27 introduce the idea that $\delta^2H$ values in precipitation are related to $\delta^{18}O$ values in a predictable way that is correlated with regional air temperature patterns. Departures from the global meteoric water line are often due to the effects of evaporation, as shown by the diverging arrow in Supp. File 1.27. Reinforcing the relationship between average air temperature and the geographic pattern of $\delta^2H$ and $\delta^{18}O$ values in river water (which ultimately comes from precipitation) are Supp. Files 1.28 and 1.29, while Supp. Files 1.30 – 1.32 go into more detail about the effects of temperature. Supp. File 1.33 again shows the worldwide distribution of $\delta^2H$ and $\delta^{18}O$ values in precipitation, and the idea that distance from the coast also influences isotope values is introduced and expanded upon in Supp. Files 1.34 – 1.37. The combined effects of temperature and distance from the coast are well illustrated by Supp. File 1.38.

In Supp. File 1.39 the concept of the water cycle is returned to, and includes the routing of water through plants as they use soil water to form new plant tissue, including fruits and vegetables. The table in Supp. File 1.40 emphasizes the amount of water in common fruits and vegetables. This is a good time to point out the fact that by harvesting fruits and vegetables and bringing them to market to be sold and later consumed, we are transporting water. The example of a tomato truck is useful in this context: a tomato truck is effectively a water truck. This is also a good opportunity to introduce the idea that irrigation is necessary to grow food crops in regions where there is not enough precipitation. The map in Supp. File 1.41 shows that in the arid western United States, much more water is used for irrigation than falls as rain or snow. This implies that ground water or river water must be transported from a wetter region to the agricultural region. This is an explicit anthropogenic alteration to the water cycle through exporting or importing water.

Finally, the concepts of the water cycle, stable isotope hydrology, and water in fruits and vegetables can be applied to the overarching research question (Supp. File 1.42): Can we use the stable isotopes of hydrogen and oxygen in water as "fingerprints" to determine the origin of fruits and vegetables? The general approach to using fruit and vegetable water to identify source regions is shown in Supp. File 1.43.

### 3.3 Computer-based laboratory exercise

The computer-based laboratory exercise was designed to reinforce the concepts covered in the introductory lecture and promote hands-on learning using fruit water stable isotope data. The laboratory exercise materials are made available as Supplementary File 2 for public, non-profit use. The laboratory exercise makes use of the IsoMAP on-line isotope mapping software and resulting data products that are publicly available under a non-commercial reuse and attribution license (http://isomap.rcac.purdue.edu:8080/gridsphere). The laboratory exercise and associated data were conceptually driven by

the research question: Can we demonstrate how to use the stable isotopes of hydrogen and oxygen in water as "fingerprints" to determine where fruits and vegetables come from?

The students are provided with detailed handouts giving step-by-step instructions for completing the laboratory exercise, which may be worked individually or in small groups. The exercise begins with a brief review of concepts covered in the introductory lecture, and the material is made available again as a reference to use in completing the exercises. The sequence of measurement activities that yields fruit water isotope data is reviewed and the method for converting the measured $\delta^2H_{vap}$ and $\delta^{18}O_{vap}$ values into $\delta^2H_{liq}$ and $\delta^{18}O_{liq}$ values is explained in the printed laboratory exercise document. The concept behind correcting the fruit water $\delta^2H_{liq}$ and $\delta^{18}O_{liq}$ values for evaporation effects is explained and illustrated with examples of using the graphical and mathematical solution methods. The students are provided with paper plots of the GMWL with measured $\delta^2H_{liq}$ and $\delta^{18}O_{liq}$ values from different kinds of fruit already plotted. They use these teaching tools to graphically estimate the source water $\delta^2H_{liq}$ and $\delta^{18}O_{liq}$ values. The student-generated estimated source water values are compiled into a master list, and can be compared across estimation methods and used in the following mapping activities.

The students are directed to a website that hosts downloadable html files (included as Supp. Files 3.1 – 3.4) that are interactive maps allowing the student to use their mouse pointer to hover over a spot on the map and the isotope values of that spot appear. These interactive maps provide a good opportunity for the students to synthesize what they have learned about what controls the spatial distribution of stable isotopes in precipitation (e.g. temperature and geographic location) as they explore the maps. The students can also use the interactive maps to quickly survey a geographic area and identify potential growing regions for each fruit.

Next, the students are asked to use the on-line mapping software at http://isomap.rcac.purdue.edu to create maps of precipitation $\delta^2H$ and $\delta^{18}O$ values for areas of the world that they find interesting, and to make maps for summer and winter seasons. This exercise illustrates and emphasizes the data and data sources that go into creating a data-rich map, such as the interactive maps the students used in the previous exercise. Then they are asked to make maps that combine the precipitation isotope maps they created with the fruit source water isotope data. These predictive maps represent the likelihood or probability that a particular pixel on the map could be the origin of the fruit water, and are displayed as color-coded "heat maps" (Fig. 5).

### 3.4 Assessment of learning effectiveness

We assessed learning outcomes with evaluation pre- and post-tests (Supp. File 4) tied to learning objectives (Table 2), as well as participant feedback surveys. The pre- and post-tests were developed by preliminary testing beforehand on an audience with similar demographics to that of the anticipated students. Problematic questions and terminology in terms of "guessability" and embedded contextual clues on the pre- and post-tests were corrected through iterative rephrasing and testing prior to administering the evaluation tests to the participant group. The pre- and post-tests consisted of identical multiple-choice questions with four candidate answers. The tests were anonymized by having each participant label their tests with a unique four-digit number (the last four digits of their phone number). With this system, there could be a

possibility to track individual learning gains, though we did not do so. Instead we assessed the learning effectiveness of the group as a whole. Calculation of statistical results was done with IBM SPSS version 23 software.

## 4 Results and Discussion

### 4.1 Isotopic analyses and food sourcing results

Table 1 lists the range of fruit water $\delta^2$H and $\delta^{18}$O values measured in this study along with their calculated source waters and their store-labelled source regions (the labelled source regions were not provided to the students until the end of the laboratory exercise as a check on their calculations). The fruit water dataset in Table 1 is used for the laboratory exercises described in Section 3.3.

Figure 3 shows example fruit water H and O stable isotope data from apricots, bananas, oranges, and tomatoes from varied source regions in North and South America. A primary observation is that the fruit waters are distinct from each other when plotted in $\delta^2$H and $\delta^{18}$O space. Given the labelled origin for each of the fruits the organization of the different fruit values on the isotope plot roughly parallels that expected based on known controls of isotopic variation in precipitation water (Figure 1). That is, bananas sourced from low-latitude Peru have higher $\delta^2$H and $\delta^{18}$O values and plot near to the warm end of the GMWL, whereas tomatoes from mid-latitude and high-elevation Colorado plot near to the cold end of the GMWL.

Another important observation from Figure 3 is that all samples within a fruit type have generally similar $\delta^2$H and $\delta^{18}$O values but are separated from each other along a trend line with a lower slope than that of the GMWL. Slopes of these trend lines differ somewhat for each fruit type. These trends lines are calculated by least-squares linear regression (shown as solid lines in Figure 3) and can be interpreted as an evaporation line characteristic of each fruit type. When these evaporation lines are projected back to their intersections with the GMWL, each fruit type's source water can be read from the plot axes (i.e. via the Graphical Method shown in Figure 1). Note that the apricots, oranges and tomatoes all plot in distinct places in Figure 3, but their evaporation lines intersect the GMWL in a similar place, yielding similar source water $\delta^2$H and $\delta^{18}$O values for these fruits near the cold end of the GMWL (Fig. 1). This similarity in source water isotope values may indicate that these fruits were grown in areas with similar climates *or* that their irrigation water was sourced from areas with similar climates.

### 4.2 Synthesis of concepts

When the known source regions (from store labelling) of the apricots, oranges and tomatoes are combined with the source water information derived from isotope measurements discussed above, several key water cycle concepts can be synthesized to provide the following teaching example. The low $\delta^2$H and $\delta^{18}$O values of the fruits indicate a cold region source for the water used to grow these crops, but none of these crops are typically considered to be sourced from cold regions. For example, the state of Florida in the USA is a well-known orange producing region, and many students will be familiar with the Florida Orange Juice trademark that is well advertised in North America. The likely solution to the

conundrum of warm region crops that appear to be grown with cold region waters is that irrigation diversions bring snowmelt-sourced water from high mountain elevations down to low valley elevations where the crops are grown. This is an example of how water can be moved across the landscape in pipes or canals for human uses rather than naturally as in rivers. In a more general sense, it is an example of how humans influence the water cycle at a large scale.

Having students utilize either their own pre-existing knowledge, or knowledge gained during the teaching activity through on-line or literature research can enable better source water interpretations of the isotope data. For example, a map of the likelihood of water source locations in North America for the tomato water data shown in Figure 4 is shown in Figure 5 (data in Figure 4 and Table 1). The red regions on the map indicate a higher probability that the calculated fruit source water $\delta^{18}O$ values match that observed in summer precipitation (June, July, August), while green to blue colors indicate

lower probability. The most immediate observation from Figure 5 is that large regions are indicated with high probability of being the tomato source area, and that no single location or region provides a unique solution.

This lack of a clear-cut geographical source solution requires the students to synthesize some of the concepts introduced in the water cycle portion of the introductory lecture and include outside information into their evaluation of the map results. For example, the prediction map in Figure 5 shows high probabilities for tomato source water in regions of

high-latitude Canada, the Laurentian Great Lakes, mountainous areas of western North America and Appalachia, as well as parts of California and Mexico. This example of a scientific analysis not yielding an unambiguous result is a useful aspect to the teaching activities. In this instance, the students must assimilate external information to determine if these are likely tomato growing regions. In practice, similar situations are common in science, when the investigator must ask themselves if their results are reasonable, and if not, what other evidence should be collected to answer the question at hand.

### 4.3 Communication of learning gains

The student group was reconvened after the three one-day activity modules were completed (stream sampling for water quality, this stable isotope hydrology activity, social science activity with surveys) and the group was asked to partition themselves into equal sized groups according to individual subject matter interest. Each group was tasked with

creating a poster presentation summarizing that day's scientific activities and findings. Mentoring of the stable isotope hydrology group by one of the authors (E.O.) during the poster creation yielded the following qualitative observations. The participants were generally in agreement on which aspects of the water cycle and stable isotope hydrology concepts to include in the introduction portion of the poster. The participants all seemed to understand how the water cycle works, and how the stable isotope signature of H and O in the fruit water could give information on where the fruit was grown. There

was also general agreement about what parts of the methods were necessary to include and that photographs and conceptual plots similar to Figures 1 and 2 would communicate what was done during the laboratory exercise. There was less agreement on what the conclusions should consist of, with roughly half of the participants feeling that a straightforward presentation of confirming the labelled origin of each food was a sufficient conclusion, while the balance of the group felt that there needed

to be a stronger tie between the food sourcing aspect and general water cycle themes. Ultimately, the poster conclusions were focused largely on the food sourcing results with minor mention of the water cycle.

The teaching activities as a whole can be viewed through the lens of Experiential Learning theory, which is based on an iterative cycle of grasping a new concept (concrete experience, and abstract conceptualization) followed by transformation of the new concept (active experimentation, and reflective observation) (Kolb et al., 2001; Kolb, 2014). The grasping phase consisted of the introductory lecture and fruit water isotopic measurement demonstration, followed by the transformation phase where students worked with data and explored results in the laboratory exercise. The poster creation session provided an extension to this iterative cycle, where the students were forced to articulate their new knowledge, and in so doing confront where the gaps were in their understanding and work with their peers and the instructor to fill those gaps.

The posters were presented to the entire workshop participant group with a 20-minute talk where each member of the poster group focused on a particular aspect with which they felt comfortable. The presentations were followed by a brief question and answer period with the audience. The presentations largely confirmed observations of learning outcomes made during the small group poster creation session, with the group again divided about how far to extend the conclusions of their findings beyond simple issues of food sourcing. The question and answer period revealed individual's nuanced strengths and weaknesses in specific subject areas. Audience questions that targeted tangential concepts from, or extensions of, the presented topic sometimes received answers that combined newly-learned vocabulary terms in nonsensical ways. This type of response was common for questions that were at the edge of newly acquired knowledge areas, and was not limited to any particular participant demographic group. These responses suggest that the students, regardless of their age, scientific background, or public speaking experience had not yet reached enough exposure and usage of the new and unfamiliar knowledge of stable isotope hydrology. Threshold concept theory (Meyer and Land, 2003; 2005) posits that unfamiliar concepts without an established analogue for the student (e.g. ranges of negative numbers, such as for stable isotope values, in which the relative magnitude of each are inverse compared to that of positive numbers) can be *troublesome knowledge* and require sustained exposure to, and usage of to surpass the threshold of familiarity. This outcome implies that the instructor or discussion leader should focus on core concepts over details or implications, and that more exercises in which the students work with the unfamiliar concepts are necessary to realize more learning gains.

However, presenting the activities described here in a one day format may be fundamentally limiting to surpassing the learning thresholds that would allow each student to transform the troublesome knowledge into tractable and readily useable concepts. A way this one-day limitation could be overcome would be for the instructor and the course content to focus on strengthening the participants' understanding of core concepts framed in the context of pre-existing analogues. For example, stable isotope values and their relative differences could be discussed in terms of which values are "higher" and which are "lower", instead of numerical (negative) values. Additionally, the instructor should train the students to rely on their solid understanding of the basic concepts to answer questions, and to understand and acknowledge the limits of their new knowledge.

**4.4 Evaluation of learning effectiveness**

The evaluation pre- and post-tests (Supp. File 4) consisted of four questions with multiple choice answers tied to the learning objectives in Table 2. Tests were scored by tallying the number of correct responses out of the total possible (e.g. $x$/4, expressed hereafter as a decimal quotient). A paired-samples $t$ test was conducted to evaluate whether participants increased their understanding of the content covered in the course. Results from 25 participants (n = 25) indicated highly significant improvement in participant scores on the concept areas as a whole (measured by total test score) from pre-test (mean (M) = 2.6, standard deviation (SD) = 1.0) to post-test (M = 3.7, SD = .5), $t(24) = -5.07$, $p < .0001$) (Table 2). Because of the small sample size, a Wilcoxon nonparametric test was also performed. Results confirmed the significant $t$ test results, $z$ = -3.67, p < .0001. These results indicate that this learning activity as a whole provided effective lessons on the basics of stable isotope hydrology and the water cycle.

A paired-samples $t$ test was also conducted to evaluate increases in participant scores by test question. Bonferroni corrections, established at an alpha level of .0125, were used in order to decrease the possibility of finding significant differences by chance (Type I error). Results from 25 participants indicated statistically significant growth in two of the items (stable isotope definitions, and distribution patterns in natural systems: Table 2, Items 2 and 3): Item 2 pre-test (M = .68, SD = .48) to post-test (M = .96, SD = .2), $t(24) = -3.06$, $p = .005$) and item 3 pre-test (M = .48, SD = .51) to post-test (M = .96, SD = .2), $t(24) = -4.71$, $p < .001$). Increases in scores were also found in the other two items (water cycle concepts, and fruit water sourcing: Table 2, Items 1 and 4), however the increase was not statistically significant: Item 1 pre-test (M = .84, SD = .37) and post-test (M = .92, SD = .28) and item 4 pre-test (M = .60, SD = .50) and post-test (M = .88, SD = .33).

Because of the small sample size, nonparametric analyses, using the Related-Samples McNemar Test were also performed. Results confirmed the significant $t$ test results for stable isotope distribution patterns (Item 3, $p = .0001$). For stable isotope definitions (Item #2), results were .0035 above significance, which is close enough to confirm the significant increase of this item from pre-test to post-test ($p = .016$). These nonparametric analyses also confirmed the non-significant $t$ test results for water cycle concepts (Item 1), and fruit water sourcing (Item 4).

The evaluation pre- and post-tests were anonymized for participant identity, and we did not follow individual participant's learning progress through the exercises. In the context of a one-day course, tracking individuals would probably be of limited value, as individualized instruction would need several sessions to implement. However, if the lessons and laboratory exercises were conducted over the course of several sessions, as would be the case in a conventional course setting, individual participant progress tracking would be beneficial and could provide improved learning outcomes for each participant.

**4.5 Participant feedback and pedagogic tailoring**

Informal, qualitative participant feedback was solicited from participants after the conclusion of the teaching activities in the form of post-participation surveys, and was combined with informal observation during the teaching

activities by one of the authors (E.O.). These results indicated that the audience demographic groups (high school students, high school teachers, undergraduate students) responded to the teaching approaches differently.

Interestingly, high school teachers *and* high school students responded to teaching approaches in similar ways, even though the high school teachers had at least an undergraduate level background in science education, and thus were more similar in background scientific education to the undergraduate students. For example, none of the high school students *or* teachers were interested in using the algebraic Simultaneous Solution approach to determining source water values, and some of each group even communicated their disinterest vocally. In contrast, some undergraduates were satisfied with the Graphical Method for finding fruit source water H and O isotope values, while others wanted to calculate more precise results using the Simultaneous Solution method.

Contrasting learning styles extended to the structure of the lessons, with commonalities again between high school teachers and students, but not between teachers and undergraduate students. One high school teacher suggested that the teaching model, "I do, We do, You do" (i.e. the instructor first demonstrates, then the instructor and students repeat the activity together, and finally the students perform the activity unguided) would be more effective. Undergraduate students were able to follow the lab exercise instructions without guidance, but asked clarifying questions. Perhaps these results are not surprising or unexpected as the teachers have become subconsciously accustomed to how they themselves construct learning activities, while the undergraduates are currently immersed in classes that utilize the same teaching approach as the "Lecture then Laboratory" model used here. These disparities in pedagogic receptiveness suggest that lesson plans and approaches need to be expressly tailored to each demographic group, and that age or education level is not a good grouping denominator.

It is also worthwhile to consider potential applications for extending the teaching activities beyond that described here as a way to increase student motivation and interest. Possibilities could include the investigation of whether fruits found at local farmer's markets are sourced as claimed by a vendor, as well as applications making use of the geographical information provided by the sourcing analysis. For example, food transportation to market may involve fossil fuel resources, and therefore the carbon cycle could be tied to the water cycle. In this way, the teaching materials discussed here provide a versatile foundation to extend or build upon.

## 5 Conclusions

We developed, described here, and provided teaching materials in the Supplementary Material for an education and engagement activity that introduces hydrologic concepts using H and O stable isotope measurements in fruit water. The fruit water isotope dataset provided in this paper is a useful dataset for teaching data analysis and interpretation. The core scientific disciplinary areas of hydrology, isotope geochemistry, and predictive mapping are introduced to the students through the teaching activities and laboratory exercises. We found that students gained effective learning outcomes as measured by assessment metrics, though feedback indicated that pedagogic tailoring may be a way to increase teaching effectiveness.

This pilot study provides an example of ways that real scientific data resulting from research activities can be incorporated into teaching activities. The study described here can be used as a foundation for further development of outreach materials that may effectively engage a range of participant groups in learning about the water cycle and the ways in which humans modify the water cycle through agricultural activity.

**Acknowledgements**

We appreciate the classroom assistance of Griffin Siebert. This research was supported by NSF EPSCoR Grant IIA 1208732 awarded to Utah State University as part of the State of Utah Research Infrastructure Improvement Award. E.O.'s preparation of this manuscript was performed under the auspices of the U.S. Department of Energy by Lawrence Livermore

National Laboratory under Contract DE-AC52-07NA27344, release number LLNL-JRNL-725565. Constructive comments by S. Male and an anonymous reviewer helped improve this paper.

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

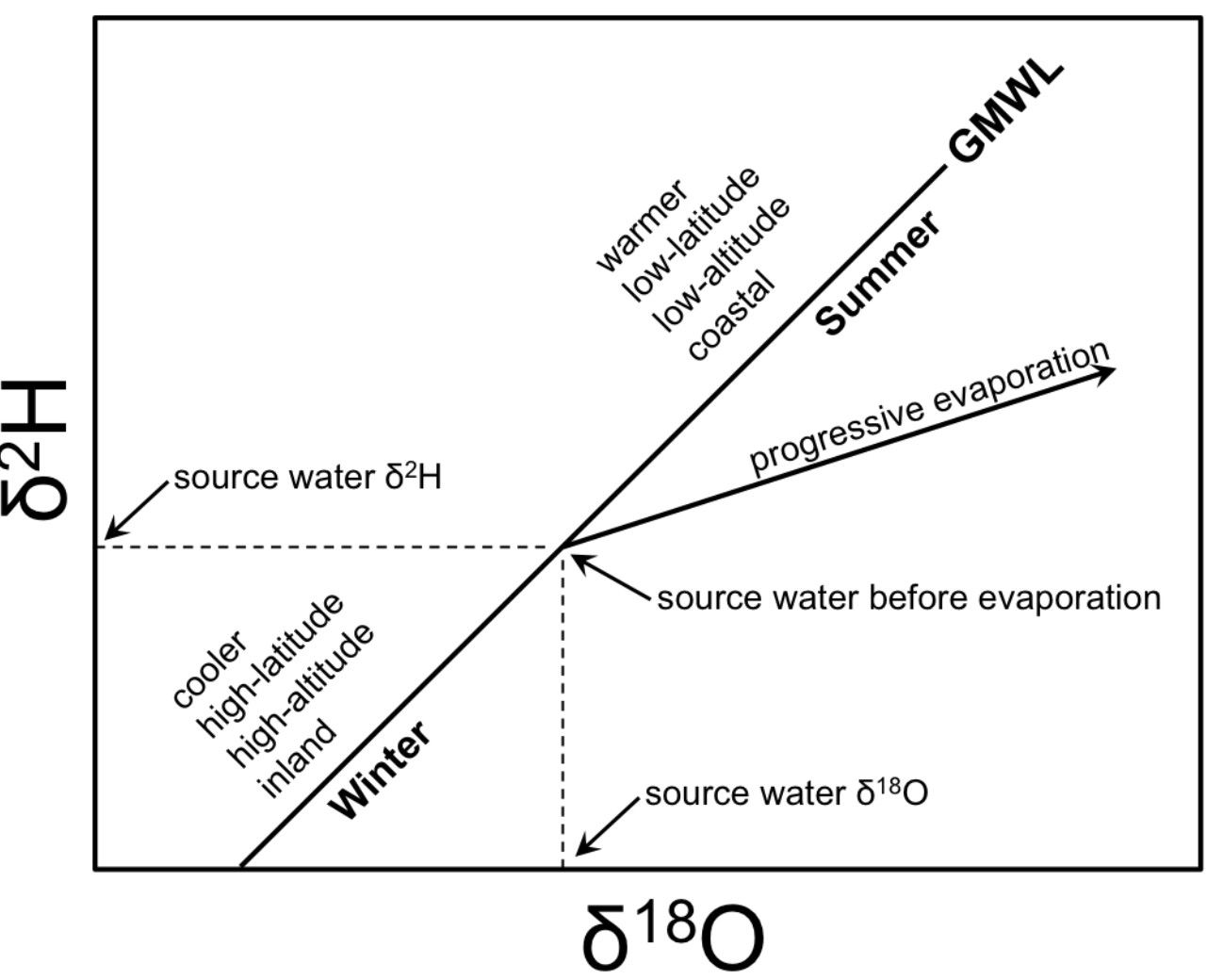

**Figure 1. The generalized relationship between $\delta^2H$ and $\delta^{18}O$ values measured in precipitation worldwide. The spatial and seasonal trends along the global meteoric water line (GMWL) are described. The evaporation line is denoted by the line labelled "progressive evaporation", and samples that have undergone more evaporation will move away from the GMWL along this line. The original $\delta^2H$ and $\delta^{18}O$ values of the evaporated water's source can be determined by finding the intersection of the evaporation line with the GMWL. Base figure redrawn from sahra.arizona.edu.**

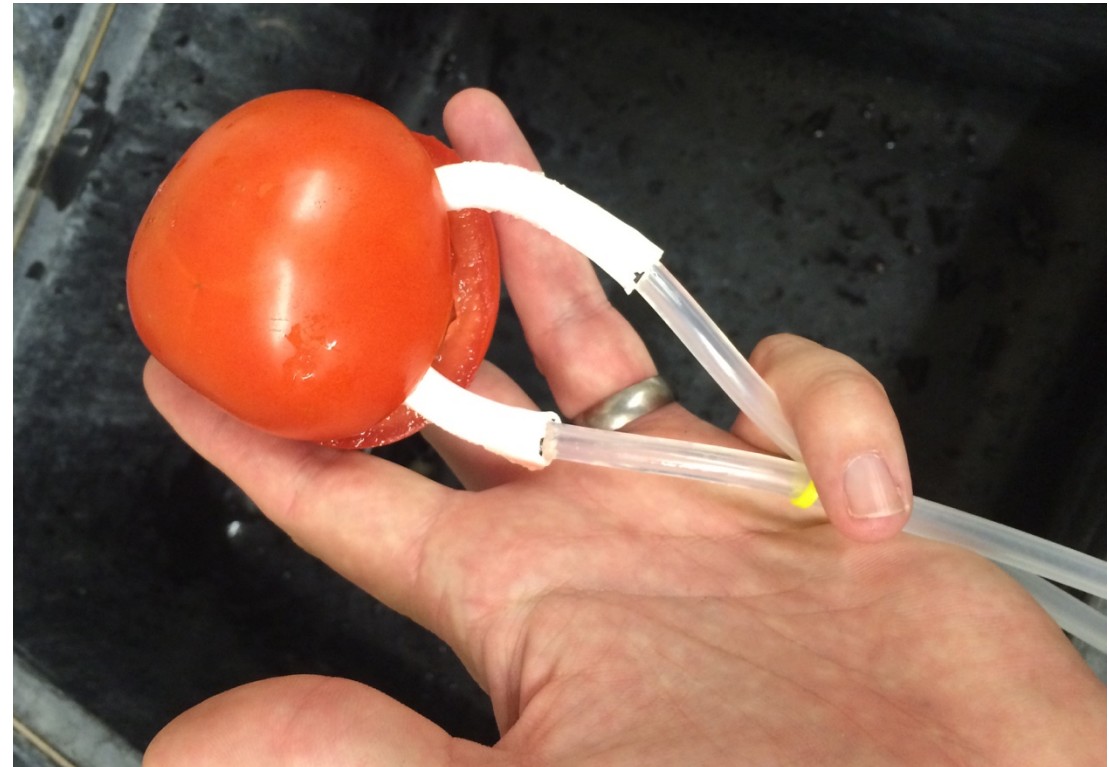

**Figure 2. Photograph of membrane-inlet probe (opaque white tubing) partially inserted into a tomato.**

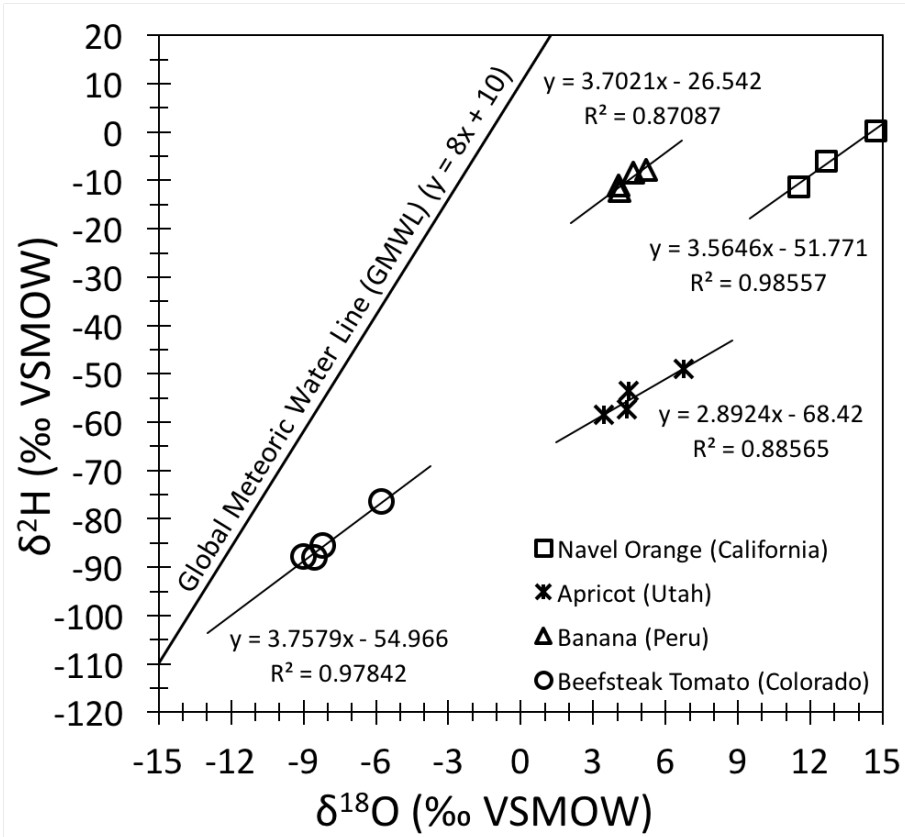

**Figure 3. Values of $\delta^2 H_{liq}$ and $\delta^{18}O_{liq}$ (symbols) calculated from measured $\delta^2 H_{vap}$ and $\delta^{18}O_{vap}$ values (not shown) for various fruits and their associated regression lines compared to the Global Meteoric Water Line (GMWL).**

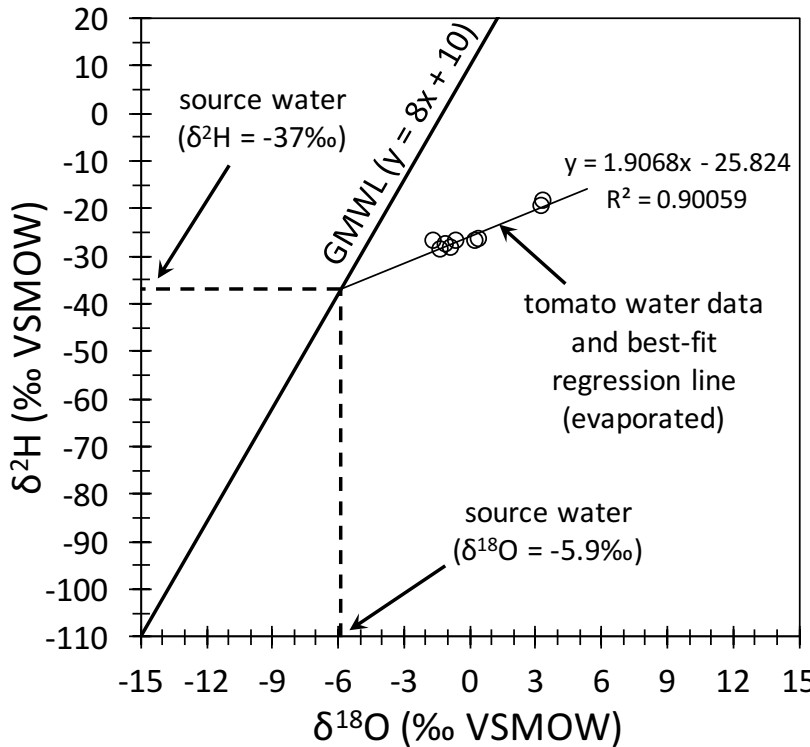

**Figure 4. Graphical method of fruit source water determination by back projecting regression line through $\delta^2H_{liq}$ and $\delta^{18}O_{liq}$ values of water inside tomatoes (open symbols) to find the source water $\delta^2H_{liq}$ and $\delta^{18}O_{liq}$ values.**

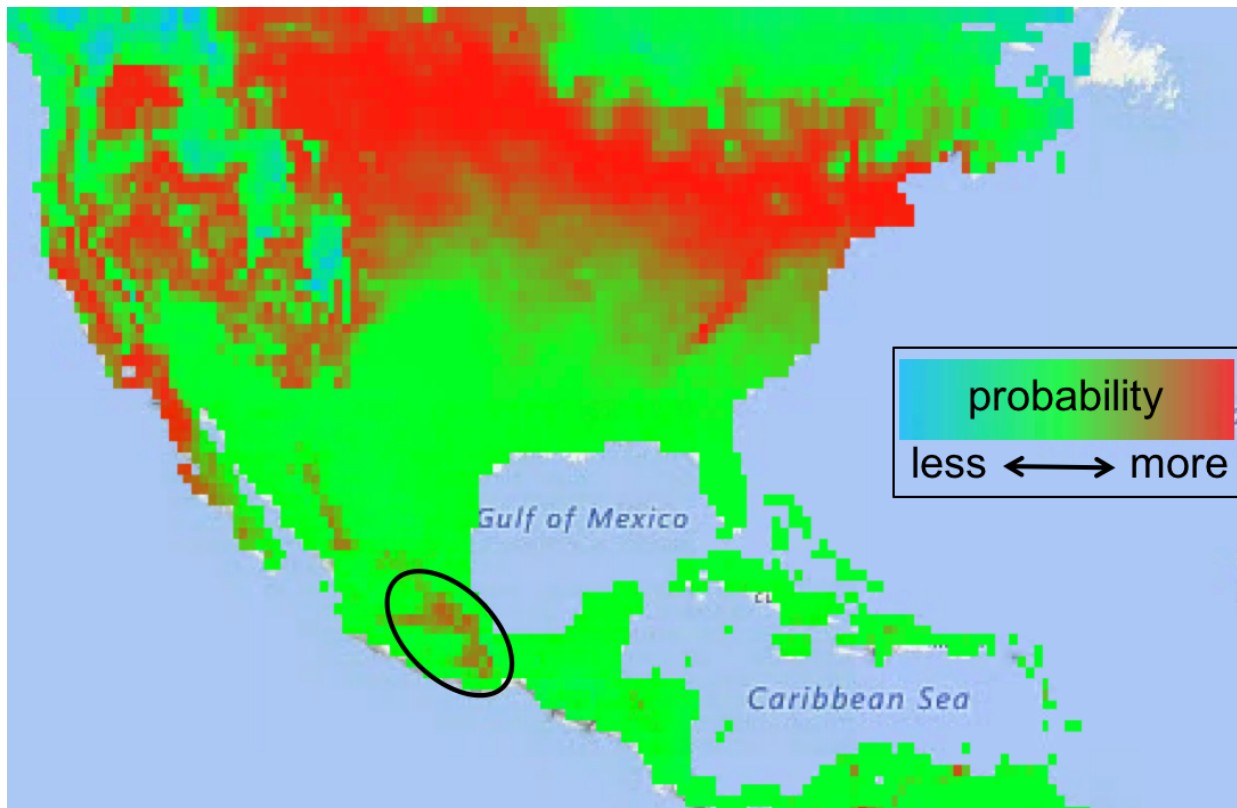

**Figure 5. Example of an assignment map created with IsoMAP software. Warm colors (red to orange) indicate higher probability that the fruit water was sourced from that location, while cooler colors (blue, exclusive of ocean area) indicate lower probability. Fruit source water $\delta^{18}O$ values used are from the tomatoes shown in Figure 4 and Table 1. Black outline over central Mexico highlights the probable source location for the tomatoes. The most immediate observation from Figure 5 is that large regions are indicated with high probability of being the tomato source area, and that no single location or region provides a unique solution (see Section 4.2 for discussion).**

| Type | Varietal | Labelled Origin | n | $\delta^{18}O_{vapor}$ | $\delta^{2}H_{vapor}$ | $\delta^{18}O_{liquid}$ | $\delta^{2}H_{liquid}$ | Regression | $\delta^{18}O_{source}$ | $\delta^{2}H_{source}$ |
|---|---|---|---|---|---|---|---|---|---|---|
| Apple | Koru | New Zealand | 4 | -11.28 | -112.9 | -1.53 | -30.8 | y = 5.1118x - 22.955 | -11.44 | -81.5 |
| | | | | (± 0.31) | (± 1.7) | (± 0.31) | (± 1.7) | $R^2 = 0.82$ | | |
| Apple | Pink Lady | Washington, USA | 4 | -17.16 | -164.4 | -7.36 | -81.7 | y = 4.7481x - 46.764 | -17.45 | -129.7 |
| | | | | (± 2.32) | (± 11.2) | (± 2.32) | (± 11.2) | $R^2 = 0.96$ | | |
| Apricot | - | Perry, Utah, USA | 4 | -5.02 | -137.3 | 4.78 | -54.6 | y = 2.8924x - 68.42 | -15.35 | -112.8 |
| | | | | (± 1.41) | (± 4.3) | (± 1.41) | (± 4.3) | $R^2 = 0.86$ | | |
| Banana | - | Peru | 4 | -5.30 | -92.6 | 4.50 | -9.9 | y = 3.7021x - 26.542 | -8.5 | -58.0 |
| | | | | (± 0.54) | (± 2.1) | (± 0.54) | (± 2.1) | $R^2 = 0.87$ | | |
| Orange | Navel | California, USA | 3 | 3.34 | -85.7 | 12.94 | -5.6 | y = 3.5646x - 51.771 | -13.93 | -101.4 |
| | | | | (± 1.62) | (± 5.8) | (± 1.62) | (± 5.8) | $R^2 = 0.99$ | | |
| Tomato | Beefsteak | Colorado, USA | 4 | -17.55 | -166.1 | -7.85 | -84.5 | y = 3.7579x - 54.966 | -15.32 | -112.5 |
| | | | | (± 1.44) | (± 5.5) | (± 1.44) | (± 5.5) | $R^2 = 0.98$ | | |
| Tomato | Hot House | Mexico | 4 | -13.46 | -129.7 | -3.86 | -49.6 | y = 2.9682x - 38.181 | -9.58 | -66.6 |
| | | | | (± 0.49) | (± 1.6) | (± 0.49) | (± 1.6) | $R^2 = 0.79$ | | |
| Tomato | Roma | Mexico | 3 | -5.42 | -109.6 | 4.28 | -28.0 | y = 2.5133x - 38.73 | -8.94 | -61.1 |
| | | | | (± 1.48) | (± 3.8) | (± 1.5) | (± 3.8) | $R^2 = 0.95$ | | |
| Tomato | On The Vine (vine #1) | USA | 3 | -9.75 | -107.0 | -0.15 | -26.9 | y = 0.4902x - 26.857 | -3.82 | -28.2 |
| | | | | (± 0.83) | (± 0.4) | (± 0.83) | (± 0.4) | $R^2 = 0.97$ | | |
| Tomato | On The Vine (vine #2) | USA | 3 | -10.52 | -108.0 | -0.92 | -27.9 | y = 2.0799x - 25.976 | -6.08 | -38.6 |
| | | | | (± 0.33) | (± 0.9) | (± 0.33) | (± 0.9) | $R^2 = 0.64$ | | |
| Tomato | On The Vine (vine #3) | USA | 3 | -7.95 | -101.6 | 1.65 | -21.5 | y = 1.6085x - 24.202 | -5.35 | -32.8 |
| | | | | (± 2.87) | (± 4.6) | (± 2.87) | (± 4.6) | $R^2 = 0.99$ | | |
| Tomato | On The Vine (all vines) | USA | 9 | -9.41 | -105.6 | 0.19 | -25.5 | y = 1.9068x - 25.824 | -5.88 | -37.0 |
| | | | | (± 1.89) | (± 3.8) | (± 1.89) | (± 3.8) | $R^2 = 0.90$ | | |
| Zucchini | - | Mexico | 4 | -12.50 | -138.9 | -2.60 | -54.0 | y = 2.5938x - 47.229 | -10.59 | -74.7 |
| | | | | (± 1.51) | (± 4.0) | (± 1.51) | (± 4.0) | $R^2 = 0.96$ | | |
| Zucchini | - | Colorado, USA | 4 | -15.23 | -149.8 | -5.23 | -64.6 | y = 2.7881x - 50.061 | -11.52 | -82.2 |
| | | | | (± 1.18) | (± 3,2) | (± 1.18) | (± 3.6) | $R^2 = 0.83$ | | |

**Table 1. Labelled origins, average measured $\delta^{2}H_{vap}$ and $\delta^{18}O_{vap}$ values, and calculated $\delta^{2}H_{liq}$ and $\delta^{18}O_{liq}$ and $\delta^{2}H_{source}$ and $\delta^{18}O_{source}$ values (± 1 St. Dev.) and regression relationships of fruits and vegetables measured in this study and used in the accompanying teaching materials.**

| Concept Area | Pre-test % correct responses | Post-test % correct responses |
|---|---|---|
| 1) Students should be able to describe how water moves through the water cycle from precipitation to soil to incorporation in plants (and fruits and vegetables). | 0.84 (± 0.37) | 0.92 (± 0.28) |
| 2) Students should be able to define what a stable isotope is. | 0.68 (± 0.48) | 0.96* (± 0.20) |
| 3) Students should be able to identify the main factor that controls how stable isotopes of water are distributed around the globe. | 0.48 (± 0.51) | 0.96** (± 0.20) |
| 4) Students should be able to explain why and how the stable isotopes in fruit and vegetabe water can yield information on the geographic origin of the food itself. | 0.6 (± 0.50) | 0.88 (± 0.38) |
| 5) Concepts as a whole | 0.65 (± 0.25) | 0.93*** (± 0.13) |

(n = 25, *$p$ = 0.005, **$p$ < 0.001, ***$p$ < 0.0001)

**Table 2. Learning objectives with pre- and post-test assessment results. Asterisks indicate level of statistical significance (*p* values) as listed below results.**

**Supplementary Material** (uploaded to manuscript website as separate files)

Supplementary Material File 1. Introductory lesson slides on stable isotope hydrology.

5    Supplementary Material File 2. Stable isotope hydrology and fruit water sourcing computer laboratory exercise.

Supplementary Material File 3. Interactive isotope maps (.html files)

        d2H_hover_Global_MJJ.html
10        d2H_hover_NorthAmerica_MJJ.html
        d18O_hover_Global_MJJ.html
        d18O_hover_NorthAmerica_MJJ.html

Supplementary Material File 4. Isotope hydrology evaluation test.

