# Peer review of "Every Apple has a Voice: Using Stable Isotopes to Teach about Food Sourcing and the Water Cycle"

_Hydrology and Earth System Sciences, 2017_

## Referee Comment (RC1) · 28 May 2017

This manuscript describes a novel approach to teach school students, school teachers, and university science students about isotope hydrology and the water cycle. The laboratory exercise developed for this teaching purpose has great promise as an interesting and relevant teaching exercise, being a forensic exercise about the origins of food - a topic of obvious relevance to everyone.

There is confusion in the manuscript, arising from insufficient clarity about three separate purposes of the study and the manuscript. You have: (1) developed modules to teach about isotope hydrology and the water cycle (2) including developing a laboratory exercise with the research question on page 5, lines 21-23 'Can we use the stable isotopes of hydrogen and oxygen in water as 'fingerprints' to determine where fruits

and vegetables come from?', and (3) evaluating the efficacy of the learning modules. A clear statement to address the following questions would have helped me. Are all three of the above purposes original and equally central to the manuscript? Is the laboratory exercise novel, or only its use as a teaching tool?

Similarly, it was perhaps not clear to the participants whether the main learning was about isotope hydrology or the water cycle. This would explain the participants' divided opinions about what should go on their posters.

My research experience is in higher education rather than isotope hydrology, and therefore my specific comments relate to these aspects of the manuscript, although much of the manuscript is about (1) and (2) above.

The design of the modules is sensible. The learning activities are consistent with the learning outcomes, and scaffolded by a presentation and demonstration.

Further detail about the data collection to evaluate the modules is necessary.

A table summarising demographic data about the participants should be added.

Early in the manuscript you report using participant feedback surveys. However on line 13 of page 11 we learn of 'informal, qualitative participant feedback'. Please explain what is meant by this. Is this the feedback surveys?

Page 2, line 1: What is meant by 'engagement'? This sentence does not make sense.

Page 2, line 9: I suggest a new paragraph here. The first 8 lines are about how important the topic is. The following lines are about to difficult it is to teach.

Page 3, line 1: Please explain the United States Next Generation Science Standards, especially for readers outside the US.

Page 4, lines 27,28: This sentence is half in the past tense and half in the present.

Page 6, lines 14 and 15 discuss a 'stumbling point'. On what basis is this considered a

stumbling point? It would be sensible to describe later in the manuscript what evidence you collected for how well your participants did or did not overcome this point.

Page 12, lines 11 and 12: Do you believe that the study was limited because you did not track individuals? What would you recommend for future research?

Page 10, line 18 refers to 'newly-learned vocabulary terms [used] in nonsensical ways'. Line 19 refers to this as a 'strategy'. Are you sure this is a strategy and not a mistake? Lines 20, 21 state that 'This outcomes implies that the instructor or discussion leader should focus on core concepts over details or implications'. On what basis do you make this conclusion? I would conclude rather that the students need more experience hearing, reading, and using the new concepts and terms. Threshold concept theory explains that when students develop understanding of troublesome concepts they start to speak like someone in the discipline. These students have not yet achieved this.

In section 4.4. mean test results are reported. Please describe the test scale. What does a mean of 2.6 indicate? Please give some examples of the test questions? Are they multiple choice, open, calculations,...?

Page 10, line 30: How do you know the lecture and demos were received uniformly well?

Overall, the initiative is exciting and could inspire many extensions. I suggest considering future possibilities for expanding the teaching modules. Beyond identifying the origins of food, students could design improvements to current practices. For example, students might consider the distances that food is transported and whether these distances could be shortened.
* * *

---

## Referee Comment (RC2) · Anonymous Referee #2 · 14 Jun 2017

General Comment: This was a really interesting approach to teaching about stable isotopes and the water cycle. The exercises and supplemental materials were sufficiently detailed and of high quality and I would consider using some of the prepared slides myself if they were available. I tried to be more critical than I am below considering that I took longer than I should have to complete this review. With that being said, I only have some minor comments that I hope are helpful to the authors.

Specific Comments: The authors write that the exercise is designed to promote "experiential learning" (page 7, line 11), and I'm curious about the motivation the students and teachers had to complete the exercises. Based on my reading of the literature, this is not experiential learning, and I would be interested to see some citations or some elaboration on why the authors think it is. Beyond that, it would be informative to know

how motivated the students and teachers were to engage in the exercise. After all, as you stated, the origin of fruits and vegetables is often indicated in the grocery store, and the isotope method does not necessarily yield a single result. In fact, the underlying question is whether or not isotopes create a "fingerprint" of origin (page 7, lines 5-6), but the answers will often be inconclusive (page 9, lines 22-27). I would caution against overselling this as experiential learning, and would suggest that this otherwise well-designed exercise could use a better framing for student engagement. For example, the isotopic fingerprint could be a method for revealing dishonest labeling practices (perhaps unscrupulous tomato vendors at a farmer's market).

On page 2, line 20, it states that revealing the geographic origin through isotope analysis makes "the distance to the point of purchase and consumption becomes more apparent." I don't know what is meant by this. Is it more apparent than looking up the origin on the sticker and then looking at a map? Also, in the same paragraph (line 23), what is the magnitude of the water flux via "trucks and trains" relative to other fluxes?

After reading the title, I thought this would be more focused on apples than it was. For example, Figure 2 is a tomato, Figs. 3 has data for oranges, apricots, bananas and tomatos. Fig. 5 if about the geographic origin of a tomato. Apples only appear in Table 1. The title seems like a bit of false advertising.

Technical Corrections: The MWL line in Fig. 1 should be labeled as GMWL because that's how you refer to it throughout the manuscript. In Fig. 3 it's labeled as the "Global Meteoric Water Line" and in Fig. 4 it's labeled as the GMWL. You should be consistent. Even in the legend of Fig. 1, you use both MWL and GMWL to refer to the same line.

The sentence on line 23-25 of page 3 is an incomplete sentence. So is this one on page 4 "These $\delta$2Hvap and $\delta$18Ovap values were then plotted in $\delta$2H and $\delta$18O space and fitting a least-squares regression line through the fruit water $\delta$2Hliq and $\delta$18Oliq values (Figure 4)."

page 4, line 13: delete the second "method" page 4, line 31: delete the "it" prior to
"the GMWL" page 6, line 11: it may not be "intuitive" to many that heavier molecules will evaporate less easily page 10, line 9-10: that last sentence is poorly phrased or incomplete. Is there a phrase missing after "largely"?
* * *

---

## Author Comment (AC1) · 17 Jun 2017

(The Author Responses to both of the Reviewer's comments below are duplicated in the attached file.)

17 June 2017

Dear HESS editor and staff,

On behalf of my coauthors, I would like to thank you for the opportunity to respond to the reviewer comments on our manuscript entitled, "Every Apple Has a Voice: Using Stable Isotopes to Teach About Food Sourcing and the Hydrologic Cycle." We also thank the reviewers for their thoughtful evaluation and comments. They bring up several key issues and points where more detail, clarification and discussion will improve the

manuscript.

Below I outline our responses to each reviewer comment:

Reviewer #1 (Sally Male):

This manuscript describes a novel approach to teach school students, school teachers, and university science students about isotope hydrology and the water cycle. The laboratory exercise developed for this teaching purpose has great promise as an interesting and relevant teaching exercise, being a forensic exercise about the origins of food - a topic of obvious relevance to everyone.

COMMENT: There is confusion in the manuscript, arising from insufficient clarity about three separate purposes of the study and the manuscript. You have: (1) developed modules to teach about isotope hydrology and the water cycle (2) including developing a laboratory exercise with the research question on page 5, lines 21-23 'Can we use the stable isotopes of hydrogen and oxygen in water as 'fingerprints' to determine where fruits and vegetables come from?', and (3) evaluating the efficacy of the learning modules. A clear statement to address the following questions would have helped me. Are all three of the above purposes original and equally central to the manuscript?

RESPONSE: This suggestion to clarify the general scope and aims of the paper and the learning exercise is a good one, and we will do so in the Introduction and Overview of Teaching Activities section (3.1), and add discussion in the Synthesis of Concepts section (4.2) and Participant feedback section (4.5).

COMMENT: Is the laboratory exercise novel, or only its use as a teaching tool?

RESPONSE: The laboratory exercise itself is novel, as this is the first application of membrane- inlet laser spectroscopy to measure fruit water isotopes in situ. Following that, its use as a teaching tool is also novel, as we do not know of another instance of this measurement system's use in teaching. We will change the language in this section to be specific about what we are claiming is novel and discuss why.

COMMENT: Similarly, it was perhaps not clear to the participants whether the main learning was about isotope hydrology or the water cycle. This would explain the participants' divided opinions about what should go on their posters.

RESPONSE: This is a good point, and may well explain the divergence in poster content at the end of the exercise. We will add some discussion of the notion that there are several main aspects of the lesson content, and that each is an important part of the whole. This relates to the prior comment about making the goals of the lesson (and this manuscript) more apparent.

COMMENT: My research experience is in higher education rather than isotope hydrology, and therefore my specific comments relate to these aspects of the manuscript, although much of the manuscript is about (1) and (2) above. The design of the modules is sensible. The learning activities are consistent with the learning outcomes, and scaffolded by a presentation and demonstration. Further detail about the data collection to evaluate the modules is necessary.

RESPONSE: We will add more detail and description of the evaluation activities to section 3.4.

COMMENT: A table summarising demographic data about the participants should be added.

RESPONSE: We will add a table as suggested to section 3.1.

COMMENT: Early in the manuscript you report using participant feedback surveys. However on line 13 of page 11 we learn of 'informal, qualitative participant feedback'. Please explain what is meant by this. Is this the feedback surveys?

RESPONSE: Yes, these were the feedback surveys as well as student feedback throughout the activities. We will add details about these surveys and the results to Section 4.5.

COMMENT: Page 2, line 1: What is meant by 'engagement'? This sentence does not

make sense.

RESPONSE: This line will be revised to clarify.

COMMENT: Page 2, line 9: I suggest a new paragraph here. The first 8 lines are about how important the topic is. The following lines are about to difficult it is to teach.

RESPONSE: We will follow this suggestion and start a new paragraph at this point.

COMMENT: Page 3, line 1: Please explain the United States Next Generation Science Standards, especially for readers outside the US.

RESPONSE: Yes, of course this is a good suggestion and we will add explanatory detail on the standards.

COMMENT: Page 4, lines 27,28: This sentence is half in the past tense and half in the present.

RESPONSE: This sentence will be corrected.

COMMENT: Page 6, lines 14 and 15 discuss a 'stumbling point'. On what basis is this considered a stumbling point? It would be sensible to describe later in the manuscript what evidence you collected for how well your participants did or did not overcome this point.

RESPONSE: We cite Orton and Forbisher (2004) as evidence that these aspects are common issues for students to overcome. Additionally, in working with the students in this exercise, there was feedback that the students did indeed have difficulty understanding the concept of stable isotope values as a ratio of ratios, as well as how to interpret values that are lower ("more negative") or higher ("less negative"). We will incorporate discussion of this feedback into Section 4.2 and 4.5. This will also contribute to addressing the issue with "informal, qualitative feedback" that this reviewer also pointed out (see above).

COMMENT: Page 12, lines 11 and 12: Do you believe that the study was limited because you did not track individuals? What would you recommend for future research?

RESPONSE: I think this comment refers to Page 8, Lines 11 and 12. I don't think tracking individuals would add significantly to the evaluation of the effectiveness of the teaching activities, especially in the context of a one day session. However, this section could benefit from additional detail and further discussion later in Section 4.4, which we will add. We could also add some discussion of what we could improve in future iterations, as well as some discussion of what further research avenues our results suggest would be worthwhile to pursue. If these activities were expanded beyond a one day exercise, then tracking individual participants could be beneficial.

COMMENT: Page 10, line 18 refers to 'newly-learned vocabulary terms [used] in non-sensical ways'. Line 19 refers to this as a 'strategy'. Are you sure this is a strategy and not a mistake? Lines 20, 21 state that 'This outcomes implies that the instructor or discussion leader should focus on core concepts over details or implications'. On what basis do you make this conclusion? I would conclude rather that the students need more experience hearing, reading, and using the new concepts and terms. Threshold concept theory explains that when students develop understanding of troublesome concepts they start to speak like someone in the discipline. These students have not yet achieved this.

RESPONSE: This is very useful input, and we appreciate the reviewer's thoughts on different interpretations of this observation. We were not familiar with Threshold Concept Theory, and we thank the reviewer for pointing this out to us. We will include some discussion of alternative interpretations.

COMMENT: In section 4.4. mean test results are reported. Please describe the test scale. What does a mean of 2.6 indicate? Please give some examples of the test questions? Are they multiple choice, open, calculations,...?

RESPONSE: We will add details of the test questions and how they were scored, perhaps in another table, or by modifying Table 2.

COMMENT: Page 10, line 30: How do you know the lecture and demos were received uniformly well?

RESPONSE: This sentence is vague and will be clarified and substantiated.

COMMENT: Overall, the initiative is exciting and could inspire many extensions. I suggest considering future possibilities for expanding the teaching modules. Beyond identifying the origins of food, students could design improvements to current practices. For example, students might consider the distances that food is transported and whether these distances could be shortened.

RESPONSE: We will incorporate some discussion of potential extensions of the teaching activities, as suggested.

Reviewer #2:

General Comment: This was a really interesting approach to teaching about stable isotopes and the water cycle. The exercises and supplemental materials were sufficiently detailed and of high quality and I would consider using some of the prepared slides myself if they were available. I tried to be more critical than I am below considering that I took longer than I should have to complete this review. With that being said, I only have some minor comments that I hope are helpful to the authors. Specific Comments:

COMMENT: The authors write that the exercise is designed to promote "experiential learning" (page 7, line 11), and I'm curious about the motivation the students and teachers had to complete the exercises. Based on my reading of the literature, this is not experiential learning, and I would be interested to see some citations or some elaboration on why the authors think it is. Beyond that, it would be informative to know how motivated the students and teachers were to engage in the exercise. After all, as you stated, the origin of fruits and vegetables is often indicated in the grocery store, and the isotope method does not necessarily yield a single result. In fact, the underlying question is whether or not isotopes create a "fingerprint" of origin (page 7,

lines 5-6), but the answers will often be inconclusive (page 9, lines 22-27). I would caution against overselling this as experiential learning, and would suggest that this otherwise well-designed exercise could use a better framing for student engagement. For example, the isotopic fingerprint could be a method for revealing dishonest labeling practices (perhaps unscrupulous tomato vendors at a farmer's market).

RESPONSE: We will reevaluate our assessment of these activities as experiential learning by exploring the relevant literature. We will also elaborate on the student's reactions to the research outcomes, in our discussion of student feedback. We do emphasize in the manuscript that the non-unique geographical solution is an important aspect, as the results are not a "Silver Bullet", as is often the case in real research. We do take the suggestion that different aspects of the results and different applications could be more interesting and effective, and we will incorporate discussion of this in the section that will pertain to future improvements and extensions (as also suggested by Reviewer #1).

COMMENT: On page 2, line 20, it states that revealing the geographic origin through isotope analysis makes "the distance to the point of purchase and consumption becomes more apparent." I don't know what is meant by this. Is it more apparent than looking up the origin on the sticker and then looking at a map?

RESPONSE: Yes, the fruits we selected to test were all labelled with origin, but many fruits in other stores are not. Yes, comparing the fruit labels to a map would provide a similar first-order result. A main point to these activities is to have the students discover something they didn't know previously, thorough their own research. This is a central concept in science, and is more thrilling than reading a label. A goal in these activities is to capture student interest and attention in a way that will be more memorable than reading a label. However, we take the suggestion to clarify these goals and our rationale for designing the activities as we did, and will include discussion of these aspects in the introduction to frame the study's goals better.

COMMENT: Also, in the same paragraph (line 23), what is the magnitude of the water flux via "trucks and trains" relative to other fluxes?

RESPONSE: We will develop quantitative estimates of these fluxes and include them in the revision.

COMMENT: After reading the title, I thought this would be more focused on apples than it was. For example, Figure 2 is a tomato, Figs. 3 has data for oranges, apricots, bananas and tomatos. Fig. 5 if about the geographic origin of a tomato. Apples only appear in Table 1. The title seems like a bit of false advertising.

RESPONSE: We respectfully disagree, and assert that the running lead of a compound title is often short and memorable, while the subtitle provides more detail in support. "Every Tomato Has a Voice," just doesn't have the same memorability as "Every Apple Has a Voice," though this may be a cultural difference. Besides, we do include data for two separate sets of apple samples. We decline to change the title, though if the Editor feels strongly about this issue, we may reconsider.

Technical Corrections:

COMMENT: The MWL line in Fig. 1 should be labeled as GMWL because that's how you refer to it throughout the manuscript. In Fig. 3 it's labeled as the "Global Meteoric Water Line" and in Fig. 4 it's labeled as the GMWL. You should be consistent. Even in the legend of Fig. 1, you use both MWL and GMWL to refer to the same line.

RESPONSE: This is indeed inconsistent and will be changed as suggested.

COMMENT: The sentence on line 23-25 of page 3 is an incomplete sentence.

RESPONSE: This sentence will be corrected.

COMMENT: So is this one on page 4 "These $\delta$2Hvap and $\delta$18Ovap values were then plotted in $\delta$2H and $\delta$18O space and fitting a least-squares regression line through the fruit water $\delta$2Hliq and $\delta$18Oliq values (Figure 4)."

RESPONSE: This sentence will be corrected.

COMMENT: page 4, line 13: delete the second "method" page 4, line 31: delete the "it" prior to C2 "the GMWL" page 6, line 11: it may not be "intuitive" to many that heavier molecules will evaporate less easily page 10, line 9-10: that last sentence is poorly phrased or incomplete. Is there a phrase missing after "largely"?

RESPONSE: These changes and clarifications will be made.

Please also note the supplement to this comment:
http://www.hydrol-earth-syst-sci-discuss.net/hess-2017-115/hess-2017-115-AC1-supplement.pdf

―――――――――――――――――

---

## Author Response (AR1)

22 June 2017

Dear HESS editor and staff,

On behalf of my coauthors, I would like to thank you for the opportunity to respond to the reviewer comments on our manuscript entitled, "Every Apple Has a Voice: Using Stable Isotopes to Teach About Food Sourcing and the Hydrologic Cycle." We also thank the reviewers for their thoughtful evaluation and comments. They bring up several key issues and points where more detail, clarification and discussion will improve the manuscript.

Below I outline our responses (in red) to each reviewer comment. Page and Line numbers of revisions refer to the marked-up manuscript version appended to the end of this response letter.

Reviewer #1 (Sally Male):

This manuscript describes a novel approach to teach school students, school teachers, and university science students about isotope hydrology and the water cycle. The laboratory exercise developed for this teaching purpose has great promise as an interesting and relevant teaching exercise, being a forensic exercise about the origins of food - a topic of obvious relevance to everyone.

COMMENT: There is confusion in the manuscript, arising from insufficient clarity about three separate purposes of the study and the manuscript. You have: (1) developed modules to teach about isotope hydrology and the water cycle (2) including developing a laboratory exercise with the research question on page 5, lines 21-23 'Can we use the stable isotopes of hydrogen and oxygen in water as 'fingerprints' to determine where fruits and vegetables come from?', and (3) evaluating the efficacy of the learning modules. A clear statement to address the following questions would have helped me.

Are all three of the above purposes original and equally central to the manuscript?

RESPONSE: Yes, these three purposes are original and equally central to the manuscript. We have the following modifications to clarify this:

P3 L8-17: In this paper, we describe how we (1) developed teaching modules about isotope hydrology and the water cycle, (2) developed novel laboratory and computer demonstrations and exercises to link water cycle components to the investigation of food sourcing, and (3) evaluate the efficacy of these teaching activities and materials. To do so, we first discuss the scientific background for this teaching activity, and provide a dataset of fruit source water H and O stable isotope values for use in the teaching activities. We then provide and discuss teaching materials in the form of lecture materials, and laboratory exercises and datasets, and provide results and discussion on the evaluation of the lesson's efficacy in achieving learning outcomes. The teaching materials are publicly available to the teaching community for non-profit use, and are included in the accompanying Supplementary Materials.

P8 L 8-10: The laboratory exercise and associated data were conceptually driven by the research question: Can we demonstrate how to use the stable isotopes of hydrogen and oxygen in water as "fingerprints" to determine where fruits and vegetables come from?

COMMENT: Is the laboratory exercise novel, or only its use as a teaching tool?

RESPONSE: We have added the following to specify what is new and novel in this manuscript:

P4 L28-29: To our knowledge, this is the first application of such a system to measure fruit water $\delta2H$ and $\delta18O$ values in situ, as well as the use of such a system as a teaching tool.

COMMENT: Similarly, it was perhaps not clear to the participants whether the main learning was about isotope hydrology or the water cycle. This would explain the participants' divided opinions about what should go on their posters.

RESPONSE: We have clarified the goals of the teaching activities as:

P3 L1-7: The goals of the teaching activities were to introduce and establish key water cycle concepts, reveal human modification of the natural water cycle, introduce core concepts of stable isotope hydrology, and link these concepts together through scientific data analysis and synthesis in the context of a food sourcing study. Additionally, because the week-long workshop was conducted as outreach from large teaching and research universities towards ambitious students and teachers, an over-arching goal of the workshop activities (and this paper) was to inspire and nurture scientific interest in the participants by exposing them to a variety of environmental science activities.

COMMENT: My research experience is in higher education rather than isotope hydrology, and therefore my specific comments relate to these aspects of the manuscript, although much of the manuscript is about (1) and (2) above.

The design of the modules is sensible. The learning activities are consistent with the learning outcomes, and scaffolded by a presentation and demonstration.

COMMENT: Further detail about the data collection to evaluate the modules is necessary.

RESPONSE: We have added the following detail about assessment:

P9 L6-11: The pre- and post-tests consisted of identical multiple-choice questions with four candidate answers. The tests were anonymized by having each participant label their tests with a unique four-digit number (the last four digits of their phone number). With this system, there could be a possibility to track individual learning gains, though we did not do so. Instead we assessed the learning effectiveness of the group as a whole.

P12 L13-15: The evaluation pre- and post-tests (Supp. File 4) consisted of four questions with

multiple choice answers tied to the learning objectives in Table 2. Tests were scored by tallying the number of correct responses out of the total possible (e.g. x/4, expressed hereafter as a decimal quotient).

COMMENT: A table summarising demographic data about the participants should be added.

RESPONSE: We do not have very much detail on the course participants beyond the general criteria with which they were selected to participate in the program. As such, an entire table is not warranted. However, we do add the detail we have:

P5 L20-24: This teaching and outreach activity took place in the context of a week-long outreach workshop for 9 high school students with an interest in pursuing environmental science, 9 undergraduate students who are actively studying environmental science and water issues, and 9 high school teachers who teach physical science subjects such as chemistry and earth.

COMMENT: Early in the manuscript you report using participant feedback surveys. However on line 13 of page 11 we learn of 'informal, qualitative participant feedback'. Please explain what is meant by this. Is this the feedback surveys?

RESPONSE: Yes, these were the feedback surveys as well as student feedback throughout the activities. We have included clarification:

P13 L10-12: Informal, qualitative participant feedback was solicited from participants after the conclusion of the teaching activities in the form of post-participation surveys, and was combined with informal observation during the teaching activities by one of the authors (E.O.).

COMMENT: Page 2, line 1: What is meant by 'engagement'? This sentence does not make sense.

RESPONSE: This line was revised:

P2 L2-3: Teaching of environmental science, and enhancing student engagement with associated disciplines is a key focus of science education efforts in the United States and worldwide (Quinn et al., 2012).

COMMENT: Page 2, line 9: I suggest a new paragraph here. The first 8 lines are about how important the topic is. The following lines are about to difficult it is to teach.

RESPONSE: We decline to make this minor stylistic change, as the resulting paragraph is truncated.

COMMENT: Page 3, line 1: Please explain the United States Next Generation Science Standards, especially for readers outside the US.

RESPONSE: We added some detail on these standards as well as how the teaching activities relate to them:

P6 L8-12:  The learning objectives are tied to the three dimensions of the Next Generation Science Standards (NGSS Lead States, 2013), which are science curriculum guidelines for grades K¬–12 in the United States. The NGSS dimensions and the teaching activity associated with each component are: (A) Disciplinary Core Ideas: The Water Cycle and stable isotope hydrology, (B) Science and Engineering Practices: data collection and analysis, (C) Crosscutting Concepts: using data and geospatial relationships to evaluate food sourcing.

COMMENT: Page 4, lines 27,28: This sentence is half in the past tense and half in the present.

RESPONSE: This sentence was corrected to:

P5 L10-11: It was necessary to measure several (> 3) individual fruits of each type in order to characterize the variation in $\delta2Hvap$ and $\delta18Ovap$ values of that fruit type's population.

COMMENT: Page 6, lines 14 and 15 discuss a 'stumbling point'. On what basis is this considered a stumbling point? It would be sensible to describe later in the manuscript what evidence you collected for how well your participants did or did not overcome this point.

RESPONSE:

P7 L9: We cite Orton and Forbisher (2004) as evidence that these aspects are common issues for students to overcome. We also point the reader to the relevant enhanced discussion in section 4.3.:

P7 L11-12: More discussion of stable isotope notation as troublesome knowledge (i.e. Meyer and Land, 2003; 2005) follows in Section 4.3.

P11 L26-34… (Sec 4.3): Audience questions that targeted tangential concepts from, or extensions of, the presented topic sometimes received answers that combined newly-learned vocabulary terms in nonsensical ways. This type of response was common for questions that were at the edge of newly acquired knowledge areas, and was not limited to any particular participant demographic group. These responses suggest that the students, regardless of their age, scientific background, or public speaking experience had not yet reached enough exposure and usage of the new and unfamiliar knowledge of stable isotope hydrology. Threshold concept theory (Meyer and Land, 2003; 2005) posits that unfamiliar concepts without an established analogue for the student (e.g. ranges of negative numbers, such as for stable isotope values, in which the relative magnitude of each are inverse compared to that of positive numbers) can be troublesome knowledge and require sustained exposure to, and usage of to surpass the threshold of familiarity.  This outcome implies that the instructor or discussion leader should focus on core concepts over details or implications, and that more exercises in which the students work with the unfamiliar concepts are necessary to realize more learning gains.

However, presenting the activities described here in a one day format may be fundamentally limiting to surpassing the learning thresholds that would allow each student to transform the

troublesome knowledge into tractable and readily useable concepts. A way this one-day limitation could be overcome would be for the instructor and the course content to focus on strengthening the participants' understanding of core concepts framed in the context of pre-existing analogues. For example, stable isotope values and their relative differences could be discussed in terms of which values are "higher" and which are "lower", instead of numerical (negative) values. Additionally, the instructor should train the students to rely on their solid understanding of the basic concepts to answer questions, and to understand and acknowledge the limits of their new knowledge.

COMMENT: Page 12, lines 11 and 12: Do you believe that the study was limited because you did not track individuals? What would you recommend for future research?

RESPONSE: I think this comment refers to Page 8, Lines 11 and 12. We have added the following details and discussion as noted:

P9 L7-10: The pre- and post-tests consisted of identical multiple-choice questions with four candidate answers. The tests were anonymized by having each participant label their tests with a unique four-digit number (the last four digits of their phone number). With this system, there could be a possibility to track individual learning gains, though we did not do so. Instead we assessed the learning effectiveness of the group as a whole

P13 L3-8: The evaluation pre- and post-tests were anonymized for participant identity, and we did not follow individual participant's learning progress through the exercises. In the context of a one-day course, tracking individuals would probably be of limited value, as individualized instruction would need several sessions to implement. However, if the lessons and laboratory exercises were conducted over the course of several sessions, as would be the case in a conventional course setting, individual participant progress tracking would be beneficial and could provide improved learning outcomes for each participant.

COMMENT: Page 10, line 18 refers to 'newly-learned vocabulary terms [used] in nonsensical ways'. Line 19 refers to this as a 'strategy'. Are you sure this is a strategy and not a mistake? Lines 20, 21 state that 'This outcomes implies that the instructor or discussion leader should focus on core concepts over details or implications'. On what basis do you make this conclusion? I would conclude rather that the students need more experience hearing, reading, and using the new concepts and terms. Threshold concept theory explains that when students develop understanding of troublesome concepts they start to speak like someone in the discipline. These students have not yet achieved this.

RESPONSE: This is very useful input, and we appreciate the reviewer's thoughts on different interpretations of this observation. We have included in Section 4.3 some discussion of Threshold Concept Theory, as well as discussion of how the activities could be improved.

P11 L13…: The teaching activities as a whole can be viewed through the lens of Experiential Learning theory, which is based on an iterative cycle of grasping a new concept (concrete experience, and abstract conceptualization) followed by transformation of the new concept (active experimentation, and reflective observation) (Kolb et al., 2001; Kolb, 2014). The grasping phase consisted of the introductory lecture and fruit water isotopic measurement demonstration, followed by the transformation phase where students worked with data and explored results in the laboratory exercise. The poster creation session provided an extension to this iterative cycle, where the students were forced to articulate their new knowledge, and in so doing confront where the gaps were in their understanding and work with their peers and the instructor to fill those gaps.

The posters were presented to the entire workshop participant group with a 20-minute talk where each member of the poster group focused on a particular aspect with which they felt comfortable. The presentations were followed by a brief question and answer period with the audience. The presentations largely confirmed observations of learning outcomes made during the small group poster creation session, with the group again divided about how far to extend the conclusions of their findings beyond simple issues of food sourcing. The question and answer period revealed individual's nuanced strengths and weaknesses in specific subject areas.

Audience questions that targeted tangential concepts from, or extensions of, the presented topic sometimes received answers that combined newly-learned vocabulary terms in nonsensical ways. This type of response was common for questions that were at the edge of newly acquired knowledge areas, and was not limited to any particular participant demographic group. These responses suggest that the students, regardless of their age, scientific background, or public speaking experience had not yet reached enough exposure and usage of the new and unfamiliar knowledge of stable isotope hydrology. Threshold concept theory (Meyer and Land, 2003; 2005) posits that unfamiliar concepts without an established analogue for the student (e.g. ranges of negative numbers, such as for stable isotope values, in which the relative magnitude of each are inverse compared to that of positive numbers) can be troublesome knowledge and require sustained exposure to, and usage of to surpass the threshold of familiarity. This outcome implies that the instructor or discussion leader should focus on core concepts over details or implications, and that more exercises in which the students work with the unfamiliar concepts are necessary to realize more learning gains.

However, presenting the activities described here in a one day format may be fundamentally limiting to surpassing the learning thresholds that would allow each student to transform the troublesome knowledge into tractable and readily useable concepts. A way this one-day limitation could be overcome would be for the instructor and the course content to focus on strengthening the participants' understanding of core concepts framed in the context of pre-existing analogues. For example, stable isotope values and their relative differences could be discussed in terms of which values are "higher" and which are "lower", instead of numerical (negative) values. Additionally, the instructor should train the students to rely on their solid understanding of the basic concepts to answer questions, and to understand and acknowledge the limits of their new knowledge.

COMMENT: In section 4.4. mean test results are reported. Please describe the test scale. What

does a mean of 2.6 indicate? Please give some examples of the test questions? Are they multiple choice, open, calculations,...?

RESPONSE: We have added details as follows:

P12 L13-15: The evaluation pre- and post-tests (Supp. File 4) consisted of four questions with multiple choice answers tied to the learning objectives in Table 2. Tests were scored by tallying the number of correct responses out of the total possible (e.g. x/4, expressed hereafter as a decimal quotient).

COMMENT: Page 10, line 30: How do you know the lecture and demos were received uniformly well?

RESPONSE: The evidence did not allow us to discriminate between assessment results pertaining to either the lecture or lab exercises separately, and therefore we are not able to support this conclusion. This sentence was removed.

COMMENT: Overall, the initiative is exciting and could inspire many extensions. I suggest considering future possibilities for expanding the teaching modules. Beyond identifying the origins of food, students could design improvements to current practices. For example, students might consider the distances that food is transported and whether these distances could be shortened.

RESPONSE: We have incorporated the following discussion of potential extensions of the teaching activities:

P14 L1-6: It is also worthwhile to consider potential applications for extending the teaching activities beyond that described here as a way to increase student motivation and interest. Possibilities could include the investigation of whether fruits found at local farmer's markets are sourced as claimed by a vendor, as well as applications making use of the geographical information provided by the sourcing analysis. For example, food transportation to market may involve fossil fuel resources, and therefore the carbon cycle could be tied to the water cycle. In this way, the teaching materials discussed here provide a versatile foundation to extend or build upon.

Reviewer #2:

General Comment: This was a really interesting approach to teaching about stable isotopes and the water cycle. The exercises and supplemental materials were sufficiently detailed and of high quality and I would consider using some of the prepared slides myself if they were available. I tried to be more critical than I am below considering that I took longer than I should have to complete this review. With that being said, I only have some minor comments that I hope are helpful to the authors.

Specific Comments:

COMMENT: The authors write that the exercise is designed to promote "experiential learning" (page 7, line 11), and I'm curious about the motivation the students and teachers had to complete the exercises. Based on my reading of the literature, this is not experiential learning, and I would be interested to see some citations or some elaboration on why the authors think it is. Beyond that, it would be informative to know how motivated the students and teachers were to engage in the exercise. After all, as you stated, the origin of fruits and vegetables is often indicated in the grocery store, and the isotope method does not necessarily yield a single result. In fact, the underlying question is whether or not isotopes create a "fingerprint" of origin (page 7, lines 5-6), but the answers will often be inconclusive (page 9, lines 22-27). I would caution against overselling this as experiential learning, and would suggest that this otherwise well-designed exercise could use a better framing for student engagement. For example, the isotopic fingerprint could be a method for revealing dishonest labeling practices (perhaps unscrupulous tomato vendors at a farmer's market).

RESPONSE:

Regarding our assessment of the teaching activities as experiential learning, we have included the following discussion in Section 4.3:

P11 L13…: The teaching activities as a whole can be viewed through the lens of Experiential Learning theory, which is based on an iterative cycle of grasping a new concept (concrete experience, and abstract conceptualization) followed by transformation of the new concept (active experimentation, and reflective observation) (Kolb et al., 2001; Kolb, 2014). The grasping phase consisted of the introductory lecture and fruit water isotopic measurement demonstration, followed by the transformation phase where students worked with data and explored results in the laboratory exercise. The poster creation session provided an extension to this iterative cycle, where the students were forced to articulate their new knowledge, and in so doing confront where the gaps were in their understanding and work with their peers and the instructor to fill those gaps.

    The posters were presented to the entire workshop participant group with a 20-minute talk where each member of the poster group focused on a particular aspect with which they felt comfortable. The presentations were followed by a brief question and answer period with the audience. The presentations largely confirmed observations of learning outcomes made during the small group poster creation session, with the group again divided about how far to extend the

conclusions of their findings beyond simple issues of food sourcing. The question and answer period revealed individual's nuanced strengths and weaknesses in specific subject areas.

Audience questions that targeted tangential concepts from, or extensions of, the presented topic sometimes received answers that combined newly-learned vocabulary terms in nonsensical ways. This type of response was common for questions that were at the edge of newly acquired knowledge areas, and was not limited to any particular participant demographic group. These responses suggest that the students, regardless of their age, scientific background, or public speaking experience had not yet reached enough exposure and usage of the new and unfamiliar knowledge of stable isotope hydrology. Threshold concept theory (Meyer and Land, 2003; 2005) posits that unfamiliar concepts without an established analogue for the student (e.g. ranges of negative numbers, such as for stable isotope values, in which the relative magnitude of each are inverse compared to that of positive numbers) can be troublesome knowledge and require sustained exposure to, and usage of to surpass the threshold of familiarity. This outcome implies that the instructor or discussion leader should focus on core concepts over details or implications, and that more exercises in which the students work with the unfamiliar concepts are necessary to realize more learning gains.

We changed "experiential" in P8 L5 "hands-on" to better describe this component of the activities.

Regarding the isotopic results, we do emphasize in the manuscript that the non-unique geographical solution is an important aspect, as the results are not a "Silver Bullet", as is often the case in real research.

P10 L25-28: This example of a scientific analysis not yielding an unambiguous result is a useful aspect to the teaching activities. In this instance, the students must assimilate external information to determine if these are likely tomato growing regions. In practice, similar situations are common in science, when the investigator must ask themselves if their results are reasonable, and if not, what other evidence should be collected to answer the question at hand.

We do take the suggestion that different aspects of the results and different applications could be more interesting and effective, and we have incorporated the following discussion of potential extensions of the teaching activities:

P14 L1-6: It is also worthwhile to consider potential applications for extending the teaching activities beyond that described here as a way to increase student motivation and interest. Possibilities could include the investigation of whether fruits found at local farmer's markets are sourced as claimed by a vendor, as well as applications making use of the geographical information provided by the sourcing analysis. For example, food transportation to market may involve fossil fuel resources, and therefore the carbon cycle could be tied to the water cycle. In

this way, the teaching materials discussed here provide a versatile foundation to extend or build upon.

COMMENT: On page 2, line 20, it states that revealing the geographic origin through isotope analysis makes "the distance to the point of purchase and consumption becomes more apparent." I don't know what is meant by this. Is it more apparent than looking up the origin on the sticker and then looking at a map?

RESPONSE: Yes, the fruits we selected to test were all labelled with origin, but many fruits in other stores are not. Yes, comparing the fruit labels to a map would provide a similar first-order result. A main point to these activities is to have the students discover something they didn't know previously, thorough their own research. This is a central concept in science, and is more thrilling than reading a label. An informal goal in these outreach activities is to capture student interest and attention in a way that will be more memorable than reading a label. To clarify some of our rationale for designing the activities as we did, we now include:

P3 L4-7: Additionally, because the week-long workshop was conducted as outreach from large teaching and research universities towards ambitious students and teachers, an over-arching goal of the workshop activities (and this paper) was to inspire and nurture scientific interest in the participants by exposing them to a variety of environmental science activities.

COMMENT: Also, in the same paragraph (line 23), what is the magnitude of the water flux via "trucks and trains" relative to other fluxes?

RESPONSE: We have included the following:

P2 L24-30: For example, in California in 2015, 2.3E+10 kg of vegetable crops were harvested and transported to market (CDFA, 2016), which represents 2.17E+10 L of water (average vegetable water content is 95% (Spungen, 2005)). While this vegetable water is two orders of magnitude smaller than the volume of water transported for urban uses in California in 2015 (7E+12 L (Mount and Hanak, 2016)), the vegetable crop water is an enormous amount of water being moved around the landscape that many people may not have considered.

COMMENT: After reading the title, I thought this would be more focused on apples than it was. For example, Figure 2 is a tomato, Figs. 3 has data for oranges, apricots, bananas and tomatos. Fig. 5 if about the geographic origin of a tomato. Apples only appear in Table 1. The title seems like a bit of false advertising.

RESPONSE: We respectfully disagree, and assert that the running lead of a compound title is often short and memorable, while the subtitle provides more detail in support. "Every Tomato Has a Voice," just doesn't have the same memorability as "Every Apple Has a Voice," though this may be a cultural difference. Besides, we do include data for two separate sets of apple samples. We decline to change the title.

Technical Corrections:

COMMENT: The MWL line in Fig. 1 should be labeled as GMWL because that's how you refer to it throughout the manuscript. In Fig. 3 it's labeled as the "Global Meteoric Water Line" and in Fig. 4 it's labeled as the GMWL. You should be consistent. Even in the legend of Fig. 1, you use both MWL and GMWL to refer to the same line.

RESPONSE: These changes have been made.

COMMENT: The sentence on line 23-25 of page 3 is an incomplete sentence.

RESPONSE: This sentence was corrected and split up to read:

P4 L5-8: Because plants take up soil water during growth, the H and O stable isotope composition ($\delta2H$ and $\delta18O$ values) of the water in the plant represents the soil water. Therefore, the $\delta2H$ and $\delta18O$ values of the plant water can be used to identify plant water sources within the soil (Dawson et al., 2002).

COMMENT: So is this one on page 4 "These $\delta2Hvap$ and $\delta18Ovap$ values were then plotted in $\delta2H$ and $\delta18O$ space and fitting a least-squares regression line through the fruit water $\delta2Hliq$ and $\delta18Oliq$ values (Figure 4)."

RESPONSE: This sentence was corrected to read:

P5 L11-13: These fruit water $\delta2Hvap$ and $\delta18Ovap$ values were then plotted in $\delta2H$ and $\delta18O$ space and a least-squares regression line was fitted through the fruit water $\delta2Hliq$ and $\delta18Oliq$ values (Figure 4).

COMMENT: page 4, line 13: delete the second "method" page 4, line 31: delete the "it" prior to C2 "the GMWL" page 6, line 11: it may not be "intuitive" to many that heavier molecules will evaporate less easily page 10, line 9-10: that last sentence is poorly phrased or incomplete. Is there a phrase missing after "largely"?

RESPONSE: The problematic words were removed from these lines and the marked-up revised manuscript denotes the changes made.

[revised manuscript text omitted]